# The Role of MER Processing Pipelines for STN Functional Identification During DBS Surgery: A Feature-Based Machine Learning Approach

**DOI:** 10.3390/bioengineering12121300

**Published:** 2025-11-26

**Authors:** Vincenzo Levi, Stefania Coelli, Chiara Gorlini, Federica Forzanini, Sara Rinaldo, Nico Golfrè Andreasi, Luigi Romito, Roberto Eleopra, Anna Maria Bianchi

**Affiliations:** 1Functional Neurosurgery Unit, Neurosurgery Department, Fondazione IRCCS Istituto Neurologico Carlo Besta, Via Giovanni Celoria 11, 20133 Milan, Italy; 2Department of Electronic, Information and Bioengineering, Politecnico di Milano, 20133 Milan, Italy; stefania.coelli@polimi.it (S.C.);; 3Parkinson and Movement Disorders Unit, Department of Clinical Neurosciences, Fondazione IRCCS Istituto Neurologico Carlo Besta, 20133 Milan, Italy; 4Fondazione IRCCS Ca’ Granda Ospedale Maggiore Policlinico, 20122 Milan, Italy

**Keywords:** deep brain stimulation (DBS), microelectrode recordings (MER), machine learning (ML), subthalamic nucleus (STN)

## Abstract

Microelectrode recording (MER) is commonly used to validate preoperative targeting during subthalamic nucleus (STN) deep brain stimulation (DBS) surgery for Parkinson’s Disease (PD). Although machine learning (ML) has been used to improve STN localization using MER data, the impact of preprocessing steps on the accuracy of classifiers has received little attention. We evaluated 24 distinct preprocessing pipelines combining four artifact removal strategies, three outlier handling methods, and optional feature normalization. The effect of each data processing procedure’s component of interest was evaluated in function of the performance obtained using three ML models. Artifact rejection methods (i.e., unsupervised variance-based algorithm (COV) and background noise estimation (BCK)), combined with optimized outlier management (i.e., statistical outlier identification per hemisphere (ORH)) consistently improved classification performance. In contrast, applying hemisphere-specific feature normalization prior to classification led to performance degradation across all metrics. SHAP (SHapley Additive exPlanations) analysis, performed to determine feature importance across pipelines, revealed stable agreement with regard to influential features across diverse preprocessing configurations. In conclusion, optimal artifact rejection and outlier treatment are essential in preprocessing MER for STN identification in DBS, whereas preliminary feature normalization strategies may impair model performance. Overall, the best classification performance was obtained by applying the Random Forest model to the dataset treated using COV artifact rejection and ORH outlier management (accuracy = 0.945). SHAP-based interpretability offers valuable guidance for refining ML pipelines. These insights can inform robust protocol development for MER-guided DBS targeting.

## 1. Introduction

Deep brain stimulation (DBS) is a safe and effective surgical treatment used to control the symptoms of Parkinson’s disease [1]. Through DBS surgery, a permanent electrode is implanted inside the brain to chronically deliver high-frequency electrical pulses to the subthalamic nucleus (STN). Although the STN is easily visible on preoperative magnetic resonance imaging (MRI), errors arising from brain shift and small uncontrollable deviations from the pre-planned electrode trajectory often require real-time intraoperative data to verify the correct placement of the electrode in the target position [2].

One method used to confirm preoperative planning is microelectrode recording (MER) [3]. MER explores the electrophysiological properties of brain tissue surrounding the STN and within the STN. In a typical DBS surgery, the microelectrodes record electrophysiological activity along a planned track as the neurosurgeon sequentially advances them into the brain. Since each part of the brain has its own characteristic neural activity (such as spike firing counts and patterns), that of the STN can be recognized over the background noise level [4]. As a result, based on the monitoring of this electrophysiological activity, the neurosurgeon decides when the microelectrode has entered the STN. However, the intraoperative interpretation of MER may be challenging and time-consuming, requiring a high level of proficiency and expertise. Recently, several machine learning (ML) algorithms have been proposed as decision-making support tools with the aim of minimizing subjectivity and improving patient outcomes [5,6].

These algorithms are based on a variety of different machine learning paradigms. Most methods use traditional approaches that classify signals based on specific features known to be of interest, such as power in particular frequency bands or various spike-dependent or spike-independent features [7,8]. Others exploit a more modern, deep learning approach in which the entire signal is provided to the algorithm rather than reducing it to a smaller vector of features [9,10,11]. One common limitation of these approaches does not lie specifically in the paradigm or ML architecture chosen, but in the methodological design of dataset processing [12,13]. Few studies indeed report in detail how the signals have been preprocessed and how the dataset has been managed [14]. Therefore, the goal of this study was to assess the impact of different MER data preprocessing and management pipelines on STN classification using feature-based ML.

Twenty-four combinations of processing approaches have been implemented, comprising four signal artifact treatments and three outlier management procedures, as well as the option to standardize or not the feature sets. Three classifiers were then trained and tested, both with and without feature selection, to evaluate the effects of the implemented pipeline on the classification results. The paper is organized as follows. In Section 2, first the analyzed dataset is described, then the implemented pipelines and the evaluation framework are explained in detail. The results are reported in Section 3 and discussed in Section 4, also addressing the study limitations. Finally, the study is summarized in Section 5.

## 2. Materials and Methods

### 2.1. Dataset

Retrospective MER data from 28 Parkinson’s disease patients undergoing bilateral DBS surgical procedures targeting the STN from 2020 to 2023 at the Neurosurgery Department of the Fondazione IRCCS Istituto Neurologico Carlo Besta (Milan, Italy) were reviewed. Patients’ mean age at surgery was 64.2 ± 9.8, with a mean disease duration of 5 years [4,9] and a male/woman ratio of 1.8. Patients signed an informed consent for the surgery procedure and data exploitation for research aims. In awake patients, MERs were collected employing a Medtronic LeadpointTM system (Medtronic, Inc., Minneapolis, MN, USA). Stereotactic planning was performed using the StealthStation Surgical Navigation System (Medtronic). Direct targeting on T2 preoperative MRI was always performed for the stereotactic targeting of the STN. All patients underwent preoperative stereotactic imaging with MRI or CT and immediate intraoperative verification of the electrode positioning with an intraoperative CT scan (O-Arm, Medtronic). The surgical procedures were performed using the Vantage frame (Elekta, Stockholm, Sweden). Glass-coated platinum/iridium microelectrodes with impedance of 0.4–1.0 MOhm (FHC, Inc., Bowdoinham, ME, USA) were used. Each hemisphere was explored using three simultaneous parallel microelectrodes (anterior, central, and posterior) spaced 2 mm apart one from another. The recordings lasted at least 10 s for each single depth. MERs started at an estimated distance from the target (EDT) of 10 mm and extended to 5 mm EDT after the planned target, advancing with 1 mm steps. Traces were high-pass filtered at 200 Hz to allow for the visualization of the firing activity during the surgery, and digitalized with a sampling rate of 24 kHz. Along the electrode path, various functional regions were identified, including white matter, thalamus, zona incerta, substantia nigra, and STN. For each patient’s surgical procedure, the intraoperative annotations were reviewed by an expert neurosurgeon and the MER traces assigned to two classes: ‘STN’ if recognized as within the STN, and ‘NOT STN’ if belonging to other structures.

### 2.2. Dataset Preparation Pipelines

In this section, the different steps adopted to compose the pipelines tested in the current work are described. The full framework is displayed in Figure 1.

#### 2.2.1. Data Preparation and Preprocessing

Data preparation steps were performed offline in the MATLAB (version R2023b) environment. First, only MER traces recorded within an EDT range [−5 mm: +2 mm] were considered to improve the dataset balance between classes (i.e., within and outside the STN). The selected raw signals underwent three artifact detection procedures: (i) visual inspection by an expert (EXP); (ii) detection via an unsupervised variance-based algorithm (COV), and (iii) detection via an algorithm based on thresholds estimated on the signal noise characteristics (BCK). Additionally, we considered the RAW dataset unprocessed for artifacts, for comparative purposes. The EXP approach consisted of an expert (V. L.) visually screening all the traces to identify evident artifacts (e.g., electromagnetic interference and mechanical electrode shift). A trace marked with artifacts was fully rejected. The remaining traces comprise the dataset ‘EXP’.

The second artifact detection approach was based on the algorithm presented in [13,14] for the identification of stationary signal segments. This was preferred over other approaches to exploit its unsupervised nature. The algorithm implementation steps were the following:Segmentation of the signal x(n) into m 0.5 s segments $\{\;x_{k};\;k\;=\;1:m\};$Compute the autocorrelation of each segment $\left\{R\left(x_{k}\right);\;k\;=\;1:m\right\}$Compute the variance of the transformed segment $\left\{v_{k}=var(R\left(x_{k}\right);\;k\;=\;1:m\right\}$;Comparison of the variances of neighboring segments by computing their distance as $d_{kl}=\frac{max(v_{k},v_{l})}{min(v_{k},v_{l})},\;with\;k=\;1:m-1;l=k+1$;Creation of a distance matrix D of all possible distances between segment pairs;The matrix elements $d_{kl}$ exceeding an experimentally identified threshold (Th = 1.8) are replaced with ones and others are replaced with zeros, and an adjacency matrix A is obtained;The resulting matrix is scanned for the longest uninterrupted segment (sequence of zeros) using a greedy algorithm.

The algorithm identifies segments that are different from the rest of the signal, marks them as artifacts, and removes them from the analyzed signal.

The last approach identifies signal segments with artifacts based on specific amplitude and frequency criteria. Therefore, two assessment steps are performed. (i) *Amplitude check:* By exploiting the definition of background noise proposed by [15,16], the amplitude background noise level is computed for the whole segment (BCKtot). Then the estimation is repeated for each 0.5 s segments (${BCK}_{k}$) separately and compared to the noise level of the complete signal. If ${BCK}_{k}>20\times BCKtot$, the k-th segment is marked as artifact. (ii) *Frequency check:* the maximum amplitude of the Fourier Transform is computed for each 0.5 s segment (${max(FFT}_{k})$). If for the k-th segment the ${max(FFT}_{k})$ is higher than 2.5 times the median across all the segments in the signal, the epoch is marked as artifact.

In the current study, both the thresholds were experimentally set, as suggested in the related literature. Signal segments marked as noisy under either criterion were removed. After applying the two automatic artifact rejection approaches to the RAW dataset, in both cases, only traces with a residual length of at least 4 s [5] were further analyzed, leading to the definition of the COV dataset and the BCK dataset, respectively, in the first and second case.

Further preprocessing steps were common for the four obtained datasets, including the selection of unique recordings distinguished by depth, electrode, and hemisphere; that is, if multiple traces shared the same depth, electrode, and hemisphere, the latest in time was selected as the one with sufficient quality satisfying the surgical equipe during the DBS exploration process. Signals were finally band-passed between 300 and 3000 Hz using a second order elliptic (zero-phase) filter.

#### 2.2.2. Feature Extraction

In line with our previous works [7,8] and the literature [17,18], 22 features were extracted from each MER trace, belonging to the time and frequency domain. Given the possible different lengths of the traces, when opportune, features were normalized with respect to the number of samples N composing the analyzed signal $x\left(n\right);n=1:N$. The list of the extracted features and the relative definitions and acronyms are reported in Table 1. The selected features have been proposed in several previous studies [8,17,18,19]. Frequency domain features were extracted from the power spectral density (PSD) computed for each trace using the Welch method, with 1 s windows (50% overlap), resulting in a spectral resolution of 1 Hz. The PSD for frequency bands below 300 Hz was derived from the mean-subtracted rectified signal, as in [20]. Subsequently, power bands were extracted from these PSD values.

#### 2.2.3. Outlier Detection and Management

Another processing step worth investigating was the impact of different treatment methods on the presence of outliers. Thus, once all the features had been extracted from the prepared datasets, they were analyzed to assess their distribution and identify outliers using three different approaches:(a)NONE—the first simple possibility is not to apply any outlier detection.(b)Outlier Rejection for Hemisphere (ORH) of each patient set, the classic approach based on feature distribution to remove samples according to lower and upper bound (interquartile range—IQR) identification with a tolerance of three [21].(c)Outlier Rejection Model (ORM) based on machine learning methodologies, i.e., the local outlier factor algorithm (LOF), an unsupervised-based algorithm which computes the local density deviation of a given data point with respect to its neighbors, applied on single patient’s data [22].

#### 2.2.4. Dataset Normalization

As the last preparation step, we considered the possibility of normalizing the extracted features to reduce patients’ variability. Indeed, we applied feature standardization to all the generated datasets based on single patient cerebral hemisphere through Min–Max scaler before the ML models training procedure [17]. As a result, a total of 24 processing pipelines were defined, each of them generating a dataset with different characteristics.

**Table 1 bioengineering-12-01300-t001:** List of extracted features.

Feature	Definition
*WL—Wave or Curve length*	$WL=\frac{1}{N}\sum_{n=1}^{N-1}\left|x\left(n\right)+x\left(n+1\right)\right|$
*ZC—Zero crossing*	The number of times the signal crosses the threshold calculated by estimating the noise level of the signal
*PKS—Peaks*	Number of positive peaks identified in a signal segment normalized for the segment length.
*MAV—Mean value of the absolute amplitude*	$MAV=\frac{1}{N}\sum_{n=1}^{N}\left|x\left(n\right)\right|$
*MED—Median value of absolute amplitude*	Middle value separating the greater and lower halves of the ordered absolute amplitude $\left|x\left(n\right)\right|$ of the trace
*TH—Signal threshold*	$TH=\frac{3}{N-1}\sqrt{\sum_{n=1}^{N}{\left(x\left(n\right)-\bar{x}\right)}^{2}}$
*Root mean square (RMS) of the signal*	$RMS=\sqrt{\;\frac{1}{N}\sum_{n=1}^{N}{x(n)}^{2}}$
*AKUR—Amplitude distribution kurtosis*	$AKUR=\frac{1}{(N-1)\sigma^{4}}\sum_{n=1}^{N}\left|x\left(n\right)-{\bar{x}}^{4}\right|$
*ASKW—Amplitude distribution skewness*	$ASKW=\frac{1}{(N-1)\sigma^{3}}\sum_{n=1}^{N}\left|x\left(n\right)-{\bar{x}}^{3}\right|$
*NL—Noise level* [15,16]	Derived from the signal’s analytic envelope
*PWRA—Averaged Power*	$PWRA=\frac{1}{N}\sum_{n=1}^{N}{x(n)}^{2}$
*ANE—Average non-linear energy* [23]	$ANE=\frac{1}{N-2}\sum_{n=2}^{N-1}\begin{array}{c} {\left[x(n\right)}^{2}-\\ x\left(n-1\right)x(n+1)] \end{array}$
*powVHFrel_1*	Relative power in the 300–1000 Hz frequency range
*powVHFrel_2*	Relative power in the 1000–2000 Hz frequency range
*powVHFrel_3*	Relative power in the 2000–3000 Hz frequency range
*powHFrel_1*	Relative power in the 70–220 Hz frequency range
*powHFrel_2*	Relative power in the 220–320 Hz frequency range
*powLFrel_1*	Relative power in the 1–4 Hz frequency range
*powLFrel_2*	Relative power in the 4–8 Hz frequency range
*powLFrel_3*	Relative power in the 8–13 Hz frequency range
*powLFrel_4*	Relative power in the 13–30 Hz frequency range
*powLFrel_5*	Relative power in the 30–70 Hz frequency range

### 2.3. Classification Models

The impact of the different pipelines described above was evaluated through the performance of supervised machine learning (ML) classification models developed in Python (version 3.11.0) using the Scikit-learn library [24]. Specifically, we employed and compared three binary classifiers: a Support Vector Machine Classification (SVC) model, an Elastic Net (EN) model, and a Random Forest (RF) model. We selected model hyperparameters a priori based on domain knowledge and default library recommendations. Specifically, the SVC model adopted a Radial Basis Function (RBF) kernel, which enables non-linear decision boundaries. Other hyperparameters, including the regularization parameter (C) and kernel coefficient (gamma), were left at default values (C = 1.0, gamma = ‘scale’). A fixed random state ensured reproducibility. The EN model is a logistic regression classifier employing elastic net regularization to combine L1 and L2 penalties, with an L1 ratio of 0.5 controlling the mix. The SAGA solver was used for efficient optimization. To ensure convergence, the maximum number of iterations was increased to 10,000. A fixed random state was also used. The RF model used default hyperparameters. Specifically, the number of estimators was set to 100, the Gini criterion was used as a measure of the split quality, the maximum depth of each tree was unrestricted, the minimum number of samples required to split an internal node was two, and the minimum number of samples needed to be at a leaf node was one. A fixed random state was used to ensure reproducibility. Strategies were incorporated into the modeling pipeline to address class imbalances, which can adversely affect the performance of classification algorithms. The dataset was partitioned into training and test subsets, and preprocessing steps were applied. These included cross-validation and normalization. Cross-validation was employed to reduce overfitting and enhance the generalization ability of the models. Specifically, we applied stratified k-fold cross-validation with shuffling, maintaining the original class distribution within each fold. The dataset was divided into k = 5 non-overlapping subsets, each approximately equal in size. Each iteration used one-fold as the test set while the remaining k − 1 folds served as the training set. This process was repeated k times, ensuring that every fold was used once as the test set. The final model performance was assessed by averaging the results across all folds. Before training, within each fold features in the training set that did not previously undergo normalization were scaled using Min–Max normalization, implemented via the ‘MinMaxScaler’ function from the Scikit-learn library (Python). The normalization parameters were derived exclusively from the training set and subsequently applied to the test data. This transformation scaled each feature to the [0, 1] range across the whole dataset. The described classifiers were applied to the full set of features after an automatic feature selection approach based on Recursive Feature Elimination (RFE) with cross-validation [25], leading to six classification models being applied at the end of each processing pipeline.

#### Performance Evaluation

The classification of each model was evaluated by computing performance metrics based on the confusion matrix and the Receiver Operating Characteristic (ROC) curve at each classification split. The computed metrics were the area under the ROC curve (AUC), classification accuracy, precision, recall, and F1 score, as reported in Equations (1)–(4).(1)$$Accuracy=\frac{True\;Positive+True\;Negative}{Total\;samples}$$(2)$$Precision=\frac{True\;Positive}{True\;Positive+False\;Positive}$$(3)$$Recall=\frac{True\;Positive}{True\;Positive+False\;Negative}$$(4)$$F1score=2\times \frac{Precision\times Recall}{Precision+Recall}$$

To further assess the effect of specific pipeline components of interest, a statistical analysis was carried out by means of ANOVA tests for each performance measure. In particular, a series of three-way ANOVA tests was implemented considering the following factors: ‘ML’ as the type of ML model employed (RF, SVC, and EN), ‘Feature Selection’, meaning the application of feature selection (Yes or No), and the specific factor of interest under hypothesis (‘DATASET’, ‘OUTLIER’, and ‘NORMALIZATION’). The main factors’ effect and interaction were considered significant with *p* < 0.05, and post hoc multiple comparison analysis was carried out with Bonferroni’s *p*-value correction. The statistical analysis was performed in the MATLAB environment.

Finally, to better understand the importance of the extracted features and their impact on classification results, the Shapley additive explanations (SHAP) method was applied and the results explored [26,27]. Specifically, features were ranked by means of their SHAP absolute value and we counted how many times each feature was ranked in the top ten positions across the pipelines and folds to define the percentage of presence across all the pipelines. The counting was kept separated for the three classifiers.

## 3. Results

In this section, we describe the results of the classification pipelines that were compared, beginning with the impact of the early preparation steps or dataset preprocessing. These steps, comprising the management of MER signal artifacts, feature extraction, and subsequent outlier management, determine twelve different datasets regarding sample numerosity on which the classification models are trained. Then, a feature normalization procedure is applied to all the resulting datasets, leading to the generation of twelve additional datasets. The classification performance of the 24 tested pipelines across the different combinations is described and commented on. Finally, an exploration of the importance of the features, performed using the SHAP approach, is also reported.

### 3.1. Preprocessing Results: Datasets’ Composition

Table 2 reports the number of samples in each dataset specifying the observations for each class (STN and NOT STN).

The datasets are the results of the different combinations of early steps procedures that were tested for dataset preparation. Thus, the original ‘RAW’ dataset is composed of 1228 observations (804 NOT STN and 424 STN); the MER signals are then screened for artifact rejection by an expert (V. L.), and the whole 10 s segments are removed if marked as ‘with artifacts’.

This procedure reduced the number of total samples to 1115 in the ‘EXP’ dataset. The RAW dataset was also processed for artifact rejection using the two automatic approaches described in Section 2.2.1. Both methods did not automatically reject the whole 10 s signal segment, but only the portion containing the artifact, and the remaining segment was further considered if with a length > 4 s. This approach allowed for less drastic dataset pruning. Indeed, the COV approach rejected 12.82% of the signal, and, in terms of whole segments, only 11 were removed. As for the BCK method, 12.47% of the signal was rejected, and 21 whole segments were removed. The overall agreement between the two algorithms reached 88.5% with a statistically significant (*p* < 0.05) Cohen’s Kappa [1] with a mean across subjects equal to 0.49 (where (0.2 ≤ k ≤ 0.4 is ‘fair agreement’, 0.41 ≤ k ≤ 0.6 is ‘moderate agreement’, and 0.61 ≤ k ≤ 1 is ‘substantial agreement’). The different formulation of the rejection criteria justifies this moderate agreement. Specifically, COV identified more ‘short’ artifacts, leading to fewer entirely rejected segments with respect to BCK. After this first step, all the features described were extracted from the remaining MER segments.

At this point, the dataset was tested for outliers by applying two approaches, ORH and ORM, and the effect of this additional procedure was tested against the possibility of not controlling for the presence of outliers (NONE). Indeed, applying the ORH methods resulted in the exclusion of 20.1% of samples from the RAW dataset, 19.9% from EXP, 15.5% from COV, and 13.5% from the BCK dataset. Applying the ORM method led to the following dataset reductions: RAW = 10.9%, EXP = 11.0%, COV = 10.8%, and BCK = 10.9%.

### 3.2. Effect of Processing Pipelines on Performance Evaluation

The described classifiers, with and without RFE feature selection step, led to six classification models being applied to the 24 datasets. Performance metrics are reported as mean and standard deviation values computed across the five cross-folds. Figure 2 shows the values of performance metrics (accuracy, F1 score, AUC, precision, and recall) across all the pipelines directly comparing the six classification models. Overall, the most critical step is the application of the feature normalization based on data from the same hemisphere of each patient.

This step aims to prevent model overfitting, but in our case, it reduces the performance of all the classifiers on all possible datasets. Indeed, accuracy values pass from a range (min–max across all the pipelines) of 0.91–0.945 for the not pre-normalized datasets to an accuracy range of 0.846–0.88 when the hemisphere-based normalization is applied. The accuracy reduction due to the normalization factor was statistically significant (*p* < 0.0001), as shown in Figure 3A. Indeed, the factor ‘NORMALIZATION’ resulted in a significant effect for all the metrics with *p* < 0.0001; thus, the analysis of the effect of ‘DATASET’ and ‘OUTLIER’ factors were investigated within the normalized and non-normalized datasets.

Analyzing the effect of the other applied procedure’ components, it can be observed that the artifacts’ management approaches alone provide a weak performance improvement, particularly for the EN and RF classification models, but the effect was never statistically significant (Figure 3B). Table 3 reports these specific results for the non-normalized dataset (mean and standard deviation) of accuracy and F1 score values. A slight improvement due to the ORH procedure was obtained for the RF classification model on the COV-treated dataset (accuracy = 0.945 (0.029), F1 score = 0.915 (0.044)), and the SVC on the RAW dataset (accuracy = 0.943 (0.015), F1 score = 0.92 (0.02)). ORM, however, appears to have no discernible impact on the results. The statistical analysis of the outlier management method effect revealed the presence of significant differences for the classification accuracy, both considering the non-normalized (*p* = 0.011) and the normalized dataset (*p* = 0.0001), as reported in Figure 3C. Specifically, a significant improvement was obtained applying ORH to the non-normalized datasets (*p* = 0.009), while ORM significantly reduced the accuracy when applied to the normalized datasets with respect to ORH (*p* = 0.008) and NONE (*p* < 0.0001) approaches. As for the analysis of the effect associated with the ML model employed, it is worth noting that the ML main factor was always significant and, overall, EN showed a lower accuracy with respect to RF (*p* < 0.0001) and SVC (*p* < 0.0001), while RF and SVC obtained not significantly different performances.

As for the F1 score, the range of values is reduced from 0.867–0.922 to 0.742–0.835 across the pipelines applied to non-normalized and normalized datasets, resulting in a statistically significant F1 score reduction (*p* < 0.0001) and a significant NORMALIZATION*ML interaction (*p* = 0.0013), as shown in the Appendix A. The analysis of the two-factor interaction revealed that when considering the non-normalized set of pipelines, EN performed the worst with respect to RF (*p* = 0.011) and SVC (*p* = 0.015), while in the case of the pipelines comprising the normalization step, RF reported the lowest F1 score, significantly lower than the SVC result (*p* = 0.001). Also, in this case, the ‘DATASET’ effect was not significant, while the effect of the outlier treatment approach was found significant for both the sets of pipelines comprising the feature normalization step or not (*p* < 0.0001). In both cases, the application of the ORM approach led to a lower F1 score than when using the ORH or not removing outlier (NONE) approaches, with *p* < 0.0001. Similar trends are observed for the AUC metrics represented in Figure 2C, with also similar statistical assessment results, as represented in the Appendix A.

Looking at more detailed metrics, such as the precision and recall shown in Figure 2D,E, it is possible to appreciate the different behavior of the classification models adopted, not clearly visible considering the previous three metrics. Specifically, while EN and SVC, with or without automatic feature selection, performed similarly with a good balance between precision and recall, the RF classifier showed a higher precision and a lower recall across all the datasets, both with and without feature selection. Statistical analysis confirmed this statement as besides the significant effect of the feature normalization (*p* < 0.0001) there also was a significant two-factor interaction (*p* = 0.003), from which it clearly emerged that RF showed the higher precision values in both the non-normalized and normalized cases (*p* < 0.0001 all cases). While also for the precision metric (Appendix A) the artifact management resulted in a not significant effect, it is interesting to note that the interaction OUTLIER*ML was significant both considering the non-normalized (*p* = 0.016) and the normalized datasets (*p* < 0.0001). In particular, the application of ORM significantly decreased the precision of the SVC and EN models but not for the RF one, with respect to the ORH and NONE cases (non-normalized datasets, SVC: ORM < NONE, *p* = 0.002; ORM < ORH, *p* = 0.0004; EN: ORM < NONE, *p* = 0.0002; ORM < ORH, *p* < 0.0001; normalized datasets, SVC: ORM < NONE, *p* < 0.0001; ORM < ORH, *p* < 0.0001; EN: ORM < NONE, *p* < 0.0001; and ORM < ORH, *p* < 0.0001). Finally, the analysis of the recall values (Appendix A) confirmed that the RF model was characterized by the lowest values across pipelines; however, this was particularly significant when considering pipelines comprising the feature normalization step, with an overall drop of recall to values from a mean (±SD) of 0.88 ± 0.04 to 0.78 ± 0.06. It is also interesting to note that, while also, in this case, the ‘DATASET’ factor was not significant, the ‘OUTLIER’ method factor was significant considering the pipelines with and without the feature normalization step based on single patient’s hemisphere data, and the OUTLIER*ML interaction was also significant (normalized: *p* = 0.007; non-normalized: *p* = 0.0008). It emerged that, first of all, the RF showed the lowest recall regardless of the outlier management method, as already noted, and that only when RF was used did ORM significantly reduce recall with respect to NONE (*p* < 0.0001) and ORH (*p* < 0.0001).

As for applying a feature selection algorithm before classification, we did not observe a significant performance improvement, as in most cases only a few features were removed, while in some cases none were (maximum number of removed features = eight). Confirming this statement, the ‘Feature Selection’ factor was never statistically significant for any performance metric. This result is probably due to the already quite small proposed set of independent features, since a preliminary analysis has been performed based on the literature and previous experience to avoid the extraction of highly correlated features [8].

### 3.3. Analysis of Feature Importance

This section presents an in-depth analysis of the importance of features across different classification models. In particular, the aim was to assess the robustness of feature importance with respect to dataset preparation by quantifying their contribution to STN classification using the SHAP approach. Features were ranked by means of their SHAP value (based on probability), and we counted how many times each feature was ranked in the first ten positions across the pipelines (five times each). Based on this count, we defined the percentage of presence across all the pipelines, considering the non-normalized and the normalized features separately. The counting was kept separate for the three classifiers, since the results were model-specific. Figure 4 shows the percentage of presence for each feature in the top-ranked positions, meaning that they contributed to the classification with high importance.

It is possible to notice that some features do not seem to contribute to the classification of the STN. This observation is coherent across classifiers and has a low impact due to the application of dataset normalization. The low-ranked features are as follows(percentage of presence <15%): ZC, AKUR, ASKW, powLFre_2–5_, and powVHFrel_1_. On the other hand, some features are among the top ranked, and show good repeatability (percentage of presence >50%), such as RMS, powHFrel_1–2_, powVHFrel_2–3_, MAV, MED, PKS, and WL.

Interestingly, RF showed different feature importance attributes with respect to SVC and EN classifiers that were, instead, quite similar to each other. Moreover, it is possible to observe an effect due to the dataset normalization. Specifically, PKS, RMS, and TH showed a reduced importance for classification, while WL, NL, and powHFrel_2_ increased their contribution; after that the hemisphere-based normalization was applied. Figure 5 shows an example of a SHAP representation obtained when the pipeline reaches the highest accuracy (COV-ORH).

From this representation, it is also possible to interpret how the features impact the result. For example, high values of RMS and MAV are positively associated with the STN (class 1), while high values of PKS and powVHFrel_3_ indicate class 0 (NOT STN).

To confirm further the repeatability of feature contributions to the classification results, the same approach was repeated to analyze the features’ importance for the models trained on a selected set of features (of variable number given the automatic approach employed). Figure 6 reports the results in terms of percentage of presence in the top ten ranked features. It is possible to confirm the previous results with an overall agreement, with a few changes in the ranking.

## 4. Discussion

Accurate identification of the STN as the target for DBS electrode implantation is fundamental to efficient neuromodulation and sufficient symptom improvement in patients with Parkinson’s disease. To this end, image-guided stereotactic trajectories are carefully planned for electrode implantation. However, to cope with potential brain shifts and discriminate between close brain structures, explorative MER signals are routinely recorded to provide surgeons with additional functional information. Indeed, information from MER can be digitally extracted and analyzed, leading to the exploitation of machine learning (ML) and deep learning (DL) approaches widely applied in the literature for target detection and to provide support for the medical team [6,28].

In this context, much attention has been paid to improving the accuracy of feature-based ML and DL algorithms by increasing the complexity of the methods [9,18]. However, few studies report in detail how the signals have been preprocessed and how the dataset has been managed. In sporadic cases, the impact of these procedures has also been discussed [13,29]. Therefore, the goal of this study was to analyze the impact of different MER data processing and management pipelines on the classification of the STN using feature-based ML. To this end, we implemented an evaluation framework to assess the impact of different choices for managing specific crucial procedural steps. Combined, these choices produced a total of 24 pipelines. Three ML models were then used for binary classification, both with and without a further feature selection step. Finally, through the application of SHAP analysis, we explored the impact of the different pipelines on feature importance and model explainability. Specifically, we first analyzed the impact of different artifacts’ rejection procedures to clean the MER signals before feature extraction. Our results suggest that slight improvements in accuracy can be observed (see Table 3), and that different artifact rejection approaches (i.e., EXP, COV [13,14], and BCK [15,16]) can similarly improve MER classification compared to not removing them at all, but this improvement did not reach statistical significance. Interestingly, the effect of this varies depending on the final classification model applied: the SVC model produced the same result for all four datasets. Following features’ extraction, three outlier management procedures were considered and their effect analyzed. Results showed that the classical ORH approach generally increased both the accuracy and F1 score when applied to three out of four datasets (i.e., RAW, EXP, and COV), while no improvement was observed if applied to BCK dataset. Instead, the ORM approach did not improve performance, and, in some cases, lower accuracy was obtained compared to the possibility of not removing outliers. These considerations also apply to pipelines, including the hemisphere-based normalization step, albeit to a lesser extent. As last a manipulation, we performed a feature normalization procedure based on data from single hemisphere as suggested in [17].

While the obvious aim of this method is to improve generalizability and avoid overfitting, in our case it led to a general decrease in performance, suggesting that the intrinsic normalization applied within model training may be enough for a single-center MER classification study.

Our final analysis regarded the explainability of the features’ contribution to classification outcomes, by ranking their SHAP values across pipelines and validation folds. From this evaluation, it was possible to observe that some features (i.e., RMS, powHFrel_1–2_, powVHFrel_2–3_, MAV, MED, PKS, and WL) greatly contributed to MER classification accuracy consistently across pipelines and classification models, while some others (i.e., ZC, AKUR, ASKW, powLFre_2–5_, and powVHFrel_1_) resulted in a non-significant contribution to STN detection. Interestingly, if we compare feature contributions between normalized and non-normalized datasets, slightly different importance among the top-ranked features can be observed, while few changes are present among the less contributing ones. These considerations are still valid considering results obtained after feature selection.

In summary, in line with the previous literature [8,17,18,30,31], our study demonstrated that, besides their simplicity, feature-based ML approaches are effective in classifying MER traces for STN detection with good accuracy (best result: COV-ORH accuracy = 0.945 (0.029)). Even so, preprocessing and data preparation approaches should be carefully selected and applied since they may differently impact classification performance, both depending on the original dataset and the classification model applied.

Although our study provides an in-depth and valuable analysis of the impact of processing pipelines on STN identification using MER traces and ML approaches, it is affected by some limitations that may have influenced results interpretation. A first limitation regards the artifact detection approaches. In our study, in fact, we only evaluated the effects in terms of dataset reduction (Table 2) and final impact on classification performance, and we could not perform an accurate analysis since the systematic and precise labeling of artifacts present in our dataset was not performed. Indeed, the EXP dataset resulted from a manual dataset pruning in the presence of evident artifacts (i.e., the whole 10 s segment is discarded without identification of artifact timing and type). For the same reason, the artifact rejection thresholds were experimentally set on our data, and a systematic optimization process as in [13] was not implemented in this work, limiting the robustness of our selected thresholds.

Further study, in line with [9,13], should be performed for a complete characterization of potential artifact-affecting MERs.

In our study data from a single center, 28 patients with Parkinson’s were included, and a stratified five cross-fold validation approach was implemented [10]. This choice limited the methods and the generalizability of the results, which is a fundamental step to further improve both ML and DL’s actual applicability in clinical practice [11]. Indeed, multicentric dataset and more flexible leave-one-patient-out validation framework would increase the generalizability of STN functional localization algorithms and pipelines.

Furthermore, it should be noted that the classification models were implemented using predefined and default hyperparameter settings to avoid introducing factors that could impact STN detection performance, as our main aim was to analyze the effect of different data preparation pipelines. Indeed, our reported accuracy aligns with the literature implementing similar approaches [6].

## 5. Conclusions

The present study provides a comprehensive comparison of MER processing and dataset management procedures in different pipelines for the functional identification of the STN. The work focused on examining and describing how different choices might affect classification performance. It also provided a potential framework for comparison and useful guidelines for designing ML algorithms for STN localization based on MER traces. Specifically, the results pointed out the need to properly identify and reject artifacts, paired with the need for appropriate outlier management. The highest accuracy values were obtained for the COV-ORH pipeline, while a pre-normalization of features based on data from a single patient and brain hemisphere led to performance degradation in our study. Finally, we used the SHAP approach to further explore feature importance, which is a fundamental step that could guide and improve the implementation of future algorithms.

## Figures and Tables

**Figure 1 bioengineering-12-01300-f001:**
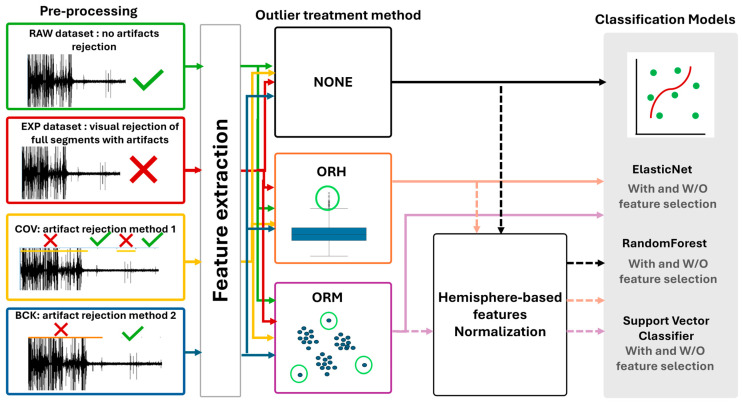
Schematic framework for comparing the implemented pipelines. Four preprocessing approaches for managing artifacts were implemented prior to feature extraction. The extracted feature sets underwent two procedures to identify and correct outliers. The twelve datasets obtained from the combinations of methods are then either passed directly to the three classification models (with or without a feature selection step) or normalized based on single patient’s hemisphere data.

**Figure 2 bioengineering-12-01300-f002:**
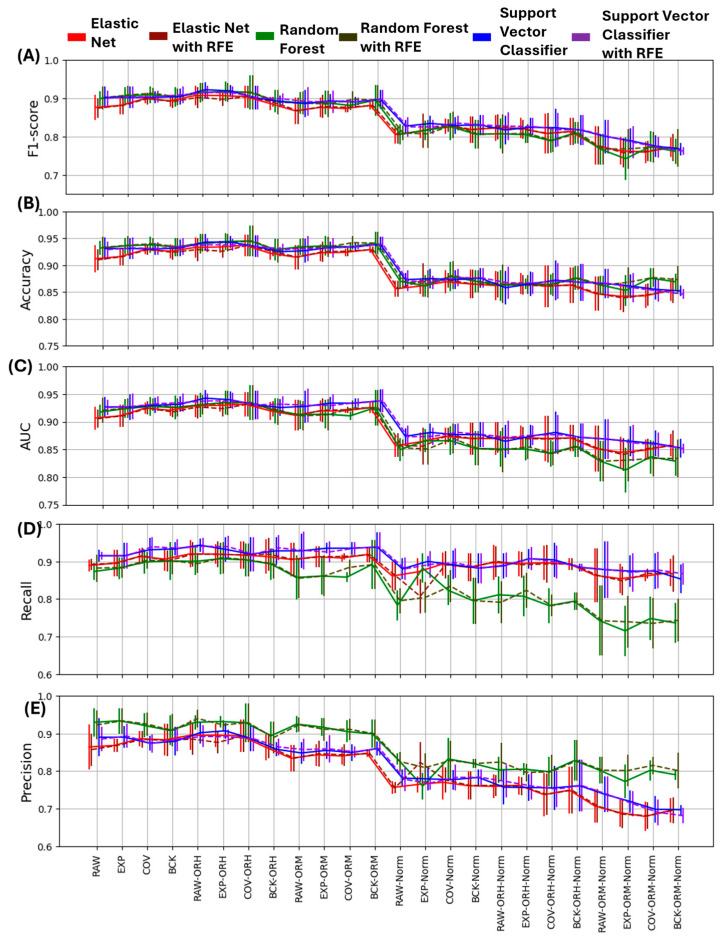
Performance evaluation metrics are shown as a function of the dataset processing pipeline. Mean values (and standard deviation) of (**A**) F1 score, (**B**) accuracy, (**C**) area under the curve (AUC), (**D**) recall, and (**E**) precision. Colors refer to different ML models and dashed lines identify models including Recursive Feature Elimination (RFE).

**Figure 3 bioengineering-12-01300-f003:**
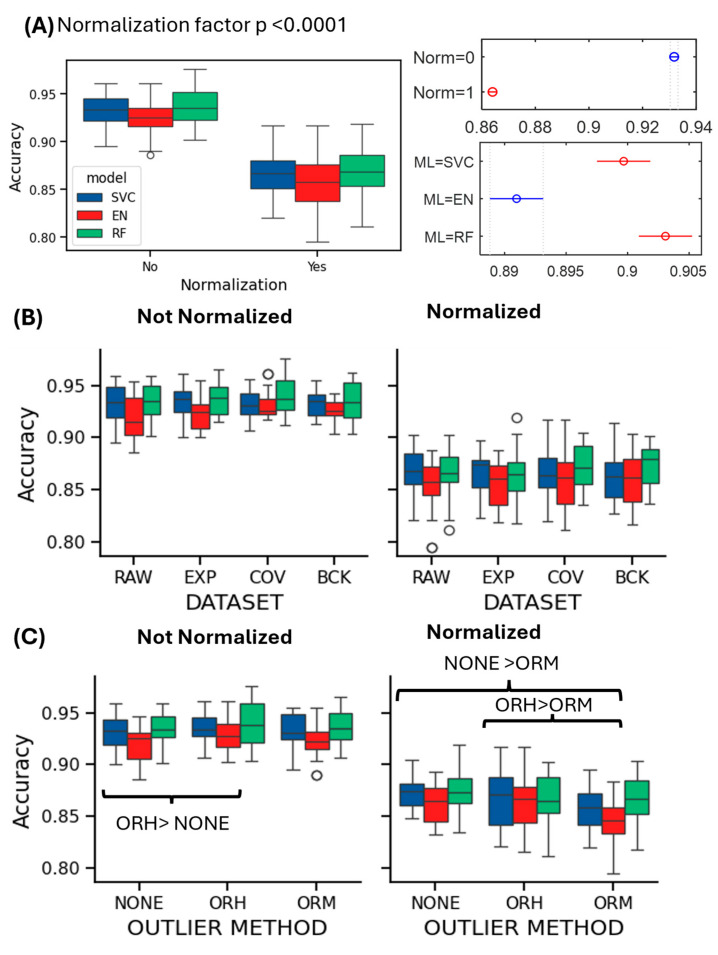
Accuracy values distributions in function of pipeline components of interest. (**A**) Accuracy values for each ML model (SVC: blue, EN: red, and RF: green) applied to the non-normalized and normalized feature datasets, (**left**) and (**right**), and the effect of the significant main factors (NORMALIZATION and ML). (**B**) Accuracy values, in function of the DATASET obtained applying different artifacts rejection approaches for each ML model for the classification of normalized and non-normalized features. (**C**) Accuracy values, in function of the OUTLIER management methods applied for each ML model for the classification of normalized and non-normalized features. Brackets identify significant differences.

**Figure 4 bioengineering-12-01300-f004:**
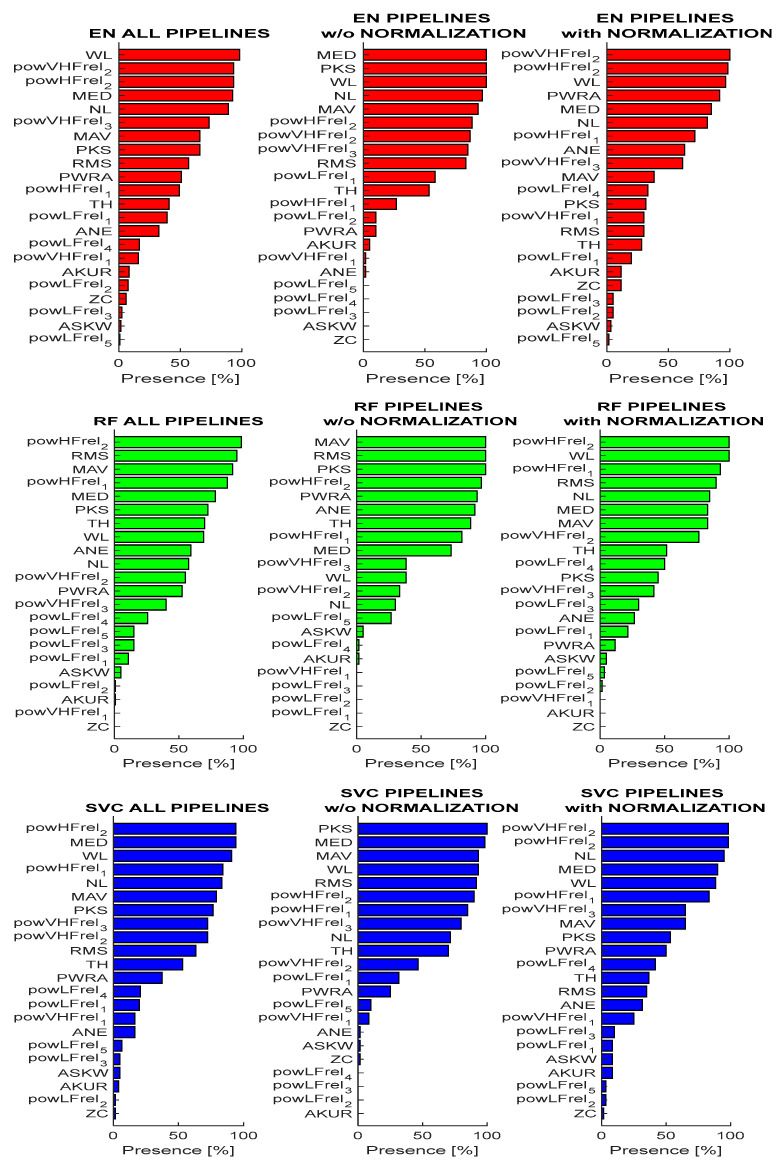
Features’ percentage of presence in the top ranked set of important features across all the pipelines, the pipeline without normalization, and the pipeline with normalization for the EN model (red), RF model (green), and SVC model (blue).

**Figure 5 bioengineering-12-01300-f005:**
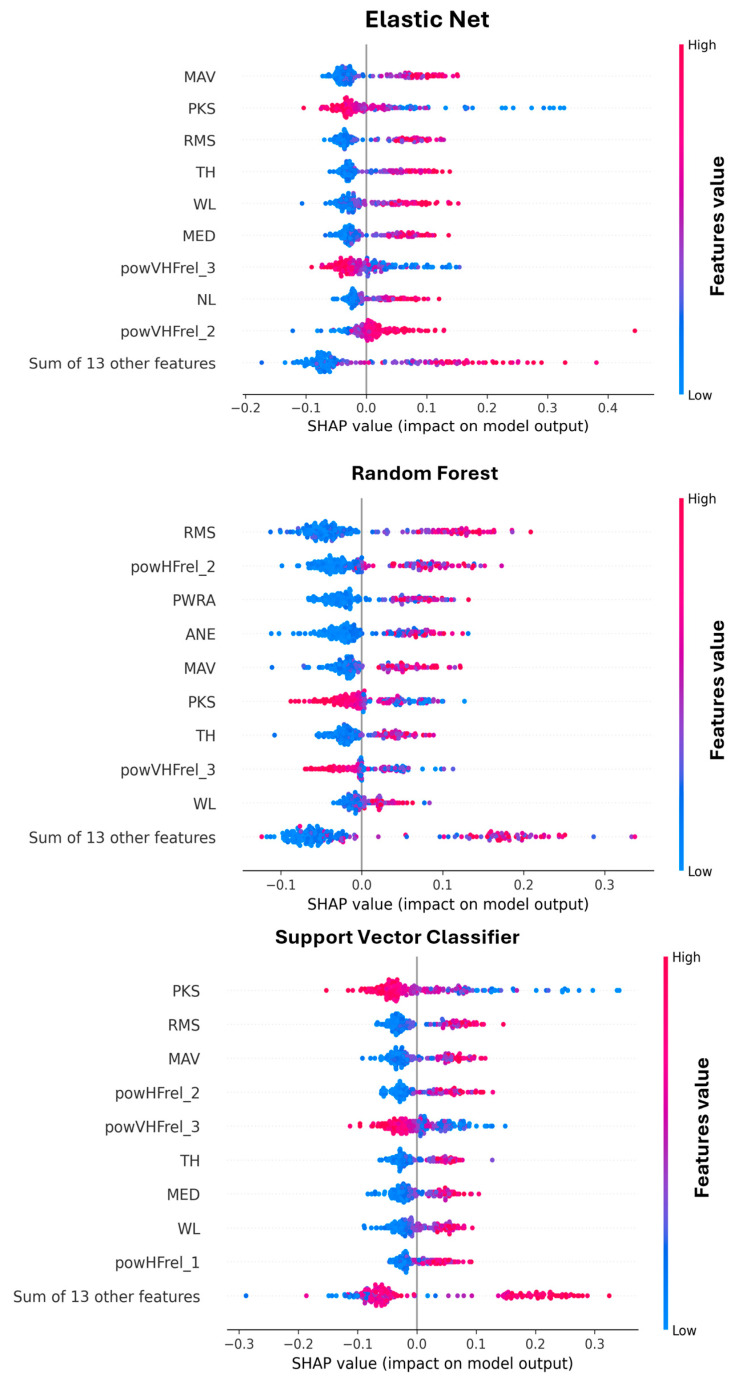
SHAP Beeswarm plot for the models applied to the pipeline reaching the highest accuracy (COV-ORH). Features’ acronyms are reported on the y-axis as presented in Table 1.

**Figure 6 bioengineering-12-01300-f006:**
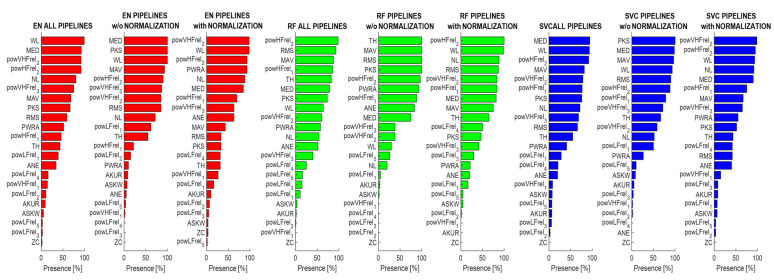
Features percentage of presence in the top-ranked set of important features across all the pipelines, the pipeline without normalization, and the pipeline with normalization for the EN model (red), RF model (green), and SVM model (blue) in the case of feature selection. Features’ acronyms are reported on the y-axis as presented in Table 1.

**Table 2 bioengineering-12-01300-t002:** Numerosity of each dataset according to the preparation pipeline. The total number of samples and the number of samples in each class (STN and NOT STN) are shown.

*Outlier* *Treatment*	*DATASET*	*RAW*	*EXP*	*COV*	*BCK*
** *NONE* **	**Total**	1228	1115	1217	1207
	**NOT STN**	804	726	794	793
	**STN**	424	389	423	414
** *ORH* **	**Total**	981	893	1028	1044
	**NOT STN**	633	582	693	699
	**STN**	348	311	335	345
** *ORM* **	**Total**	1094	992	1085	1076
	**NOT STN**	757	697	775	769
	**STN**	337	295	310	307

**Table 3 bioengineering-12-01300-t003:** Mean and standard deviation of accuracy and F1 score values obtained applying the three classifiers considering the full set of features on datasets only processed with the artifact rejection approaches.

	*EN*		*RF*		*SVC*	
	*ACC*	*F1-Score*	*ACC*	*F1-Score*	*ACC*	*F1-Score*
** *RAW* **	0.912 (0.025)	0.876 (0.032)	0.932 (0.013)	0.899 (0.019)	0.931 (0.021)	0.902 (0.028)
** *EXP* **	0.916 (0.016)	0.881 (0.023)	0.936 (0.015)	0.906 (0.022)	0.931 (0.02)	0.902 (0.028)
** *COV* **	0.929 (0.009)	0.899 (0.013)	0.938 (0.015)	0.909 (0.022)	0.93 (0.011)	0.902 (0.015)
** *BCK* **	0.926 (0.013)	0.893 (0.016)	0.934 (0.017)	0.903 (0.025)	0.932 (0.016)	0.904 (0.021)

## Data Availability

The data that support the findings of this study are available from the corresponding author, [VL], upon reasonable request.

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
