# Peer review of "The Role of MER Processing Pipelines for STN Functional Identification During DBS Surgery: A Feature-Based Machine Learning Approach"

_bioengineering, 2025, doi:10.3390/bioengineering12121300_

Round 1

Reviewer 1 Report

Comments and Suggestions for Authors

This manuscript presents a well-structured and relevant analysis of microelectrode recording (MER) preprocessing pipelines for subthalamic nucleus (STN) identification during deep brain stimulation (DBS) surgery in Parkinson’s disease. 

However, some areas are needed to be improved. 
For the Statistical validation, in this study, the author mentioned that the study reports mean ± SD but lacks significance testing between pipelines. Adding ANOVA or non-parametric tests would clarify which improvements are statistically meaningful.

Regarding to Cross-validation strategy, please consider leave-one-patient-out validation to assess inter-patient generalization.

For Normalization timing, please consider specifing whether hemisphere-based normalization occurs before or after fold splitting since pre-split normalization could cause data leakage.

For the Figures and Tables, Figures 2–5 need clearer legends and full feature names.

For the Abstract, please condense to focus on motivation, pipeline design, and key findings rather than step-by-step description.

Overall, the manuscript provides a meaningful methodological contribution and a reproducible framework that can guide future DBS-related ML studies. With statistical validation and clearer presentation, it will be suitable for publication in Bioengineering.

Comments on the Quality of English Language

The Quality of English is generally clear and comprehensible, but the manuscript would benefit from minor language editing to improve fluency and precision.

Author Response

REFEREREE 1

This paper systematically compares multiple preprocessing pipelines (24 types) for STN function identification using microelectrode recordings (MER) during DBS surgery for Parkinson's disease, and examines optimal signal processing conditions through classification accuracy and feature importance analysis using SHAP. This paper is significant in that it has a clear methodology, is highly reproducible, and quantitatively demonstrates the impact of preprocessing on machine learning models in clinical applications. With well-structured logic and appropriate use of figures and tables and SHAP analysis, it is deemed worthy of peer review.

Thank you for this comment. We report out answers in the table below

However, the following minor improvements would make the paper more persuasive.

No.

Line

Comments and Recommendations

1

L36-37

It would be good to add a sentence stating the key numerical results (e.g., Accuracy 0.945 in the COV-ORH pipeline) before "SHAP-based interpretability offers valuable guidance...".

Response:

Thank you for this suggestion, the Abstract has been improved accordingly.

2

Fig., Figure

The notation of figure is not consistent between Figure and Fig., and some are bold and some are not.

Response:

Thank you for noticing, we corrected this inconsistency

3

P.7, 9

Table 1 appears on pages 7 and 9, which is confusing. In the main text, it is written as Table, and the title of the table is TABLE.

Response:

This issue has been solved

4

L90-92

Transparency would be enhanced if simple demographic information such as age, sex, and disease duration were provided for the 28 patients (L90–92).

Response:

Thank you for this suggestion, we added these data in the method section of the paper.

5

L131, 272

The following words, "artefact" and "artifact" are mixed together (e.g. L131, L272, etc.).

Response:

We revised the test using ‘artifacts’ consistently

6

L183

The message "Error! Reference source not found" remains. This is a link error originating from Word and needs to be deleted and corrected.

Response

We apologize for this technical error, the problem has been corrected

7

L322-325

For clarity, it is helpful to reproduce the performance evaluation in Fig. 2 as explicit numerical values in the text (e.g., Accuracy 0.91–0.945).

Response:

Thank you for this suggestion. We added quantitative comments in relation to classification performance that are now stressed using statistical analysis results as requested by Referee 2.

8

Figure 4, 5

The text in Figures 4 and 5 is small and difficult to read.

Response:

Figures have been improved

9

L549

One side of the brackets in [15] in Reference is missing.

Response:

This has been corrected, thank you.

Reviewer 2 Report

Comments and Suggestions for Authors

This paper systematically compares multiple preprocessing pipelines (24 types) for STN function identification using microelectrode recordings (MER) during DBS surgery for Parkinson's disease, and examines optimal signal processing conditions through classification accuracy and feature importance analysis using SHAP. This paper is significant in that it has a clear methodology, is highly reproducible, and quantitatively demonstrates the impact of preprocessing on machine learning models in clinical applications. With well-structured logic and appropriate use of figures and tables and SHAP analysis, it is deemed worthy of peer review.

However, the minor improvements as the attached would make the paper more persuasive.

Author Response

REFEREE 2

This manuscript presents a well-structured and relevant analysis of microelectrode recording (MER) preprocessing pipelines for subthalamic nucleus (STN) identification during deep brain stimulation (DBS) surgery in Parkinson’s disease. 

Thank you for your consideration.

However, some areas are needed to be improved. 

1) For the Statistical validation, in this study, the author mentioned that the study reports mean ± SD but lacks significance testing between pipelines. Adding ANOVA or non-parametric tests would clarify which improvements are statistically meaningful.

RESPONSE: We fully agree with the Referee’s suggestion and implemented a statistical analysis based on Anova tests to statistically support our findings. To maintain sufficient statistical power, instead of comparing 24 pipelines*3 models* 2 feature selection (Yes vs No), we decided to analyze the effect of each specific factor of interest (‘DATASET’, ‘OUTLIER’ and ‘NORMALIZATION’) separately, in relation to the different models (ML) and the use or not of the feature selection procedure. In our opinion, this simplified approach allows us to better interpret the effect of each pipeline component on the classification results also in relation to specific ML model behaviors, and it could be of more interest and more informative rather than the identification of the highest performing pipeline given that it can be appreciated that performances are, in general, very similar if we consider each pipeline separately.

The statistical analysis was applied to each performance parameter, and the results are reported in the manuscript. However, we decided to graphically report only accuracy results (Figure 3) to avoid an overfilling of the manuscript, but figures representing the results for all the other parameters are reported in the supplementary document.

2) Regarding to Cross-validation strategy, please consider leave-one-patient-out validation to assess inter-patient generalization.

RESPONSE: We understand the Referee’s concern about the use of cross-validation with respect to the ‘leave-one-out’ validation approach that we also implemented in previous work. In that work, we favoured the LOO framework due to the dataset's small size. However, in the current manuscript, we opted for a stratified N-cross-fold validation approach (N = 5 in our case), as used by Maged et al. (2024) and Khorsravi et al. (2020).

As a matter of fact, the LOO validation dataset can become heavily unbalanced (e.g., a left-out patient may have recordings predominantly from one class), leading to unstable performance metrics across folds and making it difficult to compare the performance of our 24 different processing pipelines reliably. For this reason, we opted for a 5-fold sample-level stratified cross-validation. This strategy ensures that each validation fold maintains the overall class distribution (i.e., the STN-to-NO STN ratio) of the full dataset. This provides a more reliable and stable average performance metric, which was the primary goal of our methodological comparison.

However, we agree that this choice does not test for inter-patient generalisation, which represents a limitation of the current validation framework. To properly enhance and test generalisation, which is out of the scope of the current study, a multicentric dataset would be more appropriate. We acknowledged these limitations in the manuscript.

3) For Normalization timing, please consider specifing whether hemisphere-based normalization occurs before or after fold splitting since pre-split normalization could cause data leakage.

RESPONSE: The hemisphere-based normalization occurs before fold-splitting since it is patient specific and applied as suggested by Bellino et al. (2019). When the hemisphere-based normalization is not applied, a classical feature normalization is performed on the training set after fold splitting. The two approaches of normalization are, indeed, conceptually different and applied with different timing. We specified the normalization timing as required in the manuscript.

4) For the Figures and Tables, Figures 2–5 need clearer legends and full feature names.

RESPONSE: We agree that the figures need improvement. We revised the figures to improve readability. As for the full names of the features, we reported the acronyms as defined in Table I, to avoid including too much text and reducing readability.

5) For the Abstract, please condense to focus on motivation, pipeline design, and key findings rather than step-by-step description.

RESPONSE: Thank you for the suggestion, the abstract has been revised.

Overall, the manuscript provides a meaningful methodological contribution and a reproducible framework that can guide future DBS-related ML studies. With statistical validation and clearer presentation, it will be suitable for publication in Bioengineering.

Thank you for this comment and for the above suggestion.

Round 2

Reviewer 1 Report

Comments and Suggestions for Authors

Based on this revision, the authors have substantially improved the manuscript in response to the first-round review. The revised version now demonstrates strong methodological clarity, better statistical validation, and improved presentation quality.

From the structure, the cross-validation and normalization procedures are now much clearer, and the explanation for the performance degradation under hemisphere-based normalization is convincing. Moreover, Figures and tables have been upgraded for clarity, and Table I now includes units and corrected formulas. Nevertheless, the discussion more effectively links model interpretability to clinical decision-making, highlighting the relevance for DBS surgical support.

However, there are some points that still need to be fixed before publishing.

First of all, please consider adding a short supplementary analysis or statement on robustness of artifact-rejection thresholds. Moreover, please double check and ensure consistent formatting of variables (italic) and operators (roman) across all tables and captions.

For the language issues, it almost fixed. Please consider a light language polish that could further enhance flow.

Overall, this revision resolves the earlier methodological and presentation concerns. The study now stands as a methodologically sound, reproducible, and clinically relevant contribution to MER-based DBS research. Please review the suggestions mentioned above. With these suggestions, i believe that the paper would be ready for publishing.

Author Response

Dear Editor and Referees,

Thank you again for considering our manuscript and taking the time to revise it, providing us with useful suggestions and positive comments. Here we provide a point-by-point response to each comment from the Referee. Corrections in the manuscript are reported in red

Reviewer: Based on this revision, the authors have substantially improved the manuscript in response to the first-round review. The revised version now demonstrates strong methodological clarity, better statistical validation, and improved presentation quality. From the structure, the cross-validation and normalization procedures are now much clearer, and the explanation for the performance degradation under hemisphere-based normalization is convincing. Moreover, Figures and tables have been upgraded for clarity, and Table I now includes units and corrected formulas. Nevertheless, the discussion more effectively links model interpretability to clinical decision-making, highlighting the relevance for DBS surgical support.

Response: Thank you very much for this comment.

Reviewer: However, there are some points that still need to be fixed before publishing. First of all, please consider adding a short supplementary analysis or statement on robustness of artifact-rejection thresholds.

Response: We agree with the Referee that a more in-depth and systematic analysis of the artifact rejection methods would be necessary. However, as we stated in the manuscript, since our dataset is not labelled for artifact, this is, at the moment, unfeasible.  Thus, as suggested by the reviewer we opted to add a statement about the robustness of artifact-rejection thresholds in the discussion:

"A first limitation regards the artifact detection approaches. In our study, in fact, we only evaluated the effects in terms of dataset reduction (TABLE II) and final impact on classification performance, while we could not perform an accurate analysis since systematic and precise labeling of artifacts present in our dataset was not performed. Indeed, the EXP dataset resulted from a manual dataset pruning in the presence of evident artifacts (i.e., the whole 10-second segment is discarded without identification of artifact timing and type). For the same reason, the artifact rejection thresholds were experimentally set on our data, and a systematic optimization process as in [13] was not implemented in this work, limiting the robustness of our selected thresholds."

Reviewer: Moreover, please double check and ensure consistent formatting of variables (italic) and operators (roman) across all tables and captions.

Response: Thank you, we carefully checked the consistency

Reviewer: For the language issues, it almost fixed. Please consider a light language polish that could further enhance flow.

Response: Thank you, we revised it again. Corrections are highlighted in using red.

Reviewer: Overall, this revision resolves the earlier methodological and presentation concerns. The study now stands as a methodologically sound, reproducible, and clinically relevant contribution to MER-based DBS research. Please review the suggestions mentioned above. With these suggestions, i believe that the paper would be ready for publishing.

Response: Thank you for all the suggestions.
